# Oral Antiseptics against SARS-CoV-2: A Literature Review

**DOI:** 10.3390/ijerph19148768

**Published:** 2022-07-19

**Authors:** Cristian Gabriel Guerrero Bernal, Emmanuel Reyes Uribe, Joel Salazar Flores, Juan José Varela Hernández, Juan Ramón Gómez-Sandoval, Silvia Yolanda Martínez Salazar, Adrián Fernando Gutiérrez Maldonado, Jacobo Aguilar Martínez, Sarah Monserrat Lomelí Martínez

**Affiliations:** 1Especialidad de Periodoncia, Departamento de Clínicas Odontológicas Integrales, Centro Universitario de Ciencias de la Salud, Universidad de Guadalajara, Guadalajara 44340, Mexico; cristian.guerrero1532@alumnos.udg.mx (C.G.G.B.); juan.ramongom@academicos.udg.mx (J.R.G.-S.); 2Departamento de Ciencias Médicas y de la Vida, Centro Universitario de la Ciénega, Universidad de Guadalajara, Guadalajara 47810, Mexico; emmanuel.reyes@academicos.udg.mx (E.R.U.); joelos12@hotmail.com (J.S.F.); juan.varela@academicos.udg.mx (J.J.V.H.); syms@cuci.udg.mx (S.Y.M.S.); fher_gutz@hotmail.com (A.F.G.M.); 3Instituto de Investigación en Odontología, Departamento de Clínicas Odontológicas Integrales, Centro Universitario de Ciencias de la Salud, Universidad de Guadalajara, Guadalajara 44340, Mexico; 4Departamento de Ciencias Tecnológicas, Centro Universitario de la Ciénega, Universidad de Guadalajara, Guadalajara 44430, Mexico; jax781023@hotmail.com; 5Maestría en Salud Pública, Departamento de Bienestar y Desarrollo Sustentable, Centro Universitario del Norte, Universidad de Guadalajara, Guadalajara 44430, Mexico; 6Especialidad de Prostodoncia, Departamento de Clínicas Odontológicas Integrales, Centro Universitario de Ciencias de la Salud, Universidad de Guadalajara, Guadalajara 44430, Mexico

**Keywords:** COVID-19, SARS-CoV-2, oral antiseptic, infection control

## Abstract

Dentists are health care workers with the highest risk of exposure to COVID-19, because the oral cavity is considered to be a reservoir for SARS-CoV-2 transmission. The identification of SARS-CoV-2 in saliva, the generation of aerosols, and the proximity to patients during dental procedures are conditions that have led to these health care workers implementing additional disinfection strategies for their protection. Oral antiseptics are widely used chemical substances due to their ability to reduce the number of microorganisms. Although there is still no evidence that they can prevent the transmission of SARS-CoV-2, some preoperative oral antiseptics have been recommended as control measures, by different health institutions worldwide, to reduce the number of microorganisms in aerosols and droplets during dental procedures. Therefore, this review presents the current recommendations for the use of oral antiseptics against SARS-CoV-2 and analyzes the different oral antiseptic options used in dentistry.

## 1. Introduction

Coronavirus disease 19 (COVID-19), responsible for the pandemic that began in late 2019 in Wuhan, China, is caused by a virus that originated as a new type-2 severe acute respiratory syndrome coronavirus (SARS-CoV-2), and there have been more than 536 million confirmed cases up to June 2022 [1,2]; the main transmission route is through direct contact or the inhalation of aerosols [3]. The Occupational Safety and Health Administration (OSHA) of the United States Department of Labor established an occupational risk pyramid for COVID-19, which is structured into four levels of risk exposure: very high, high, medium, and low. Health workers are considered a group with a very high risk of exposure, because their procedures can generate aerosols; this particularly includes dentists, due to the close contact with patients and the instruments used [4].

Dentists present the greatest risk of exposure to COVID-19 infection. The dental devices and instruments utilized, such as ultrasonic devices, implant and surgical motors, hand pieces, and triple syringes, can generate large amounts of aerosols, which can disperse numerous bacteria and viruses [5]. All of this has prompted dental care workers to pay more attention to their protective barriers against SARS-CoV-2, including larger particle filtration masks, caps, respirators, and face shields [6,7]. However, despite these additional measures, the presence of SARS-CoV-2 has been identified in different types of materials, remaining in plastic for up to 4 days, and 7 days in stainless-steel and surgical masks [8].

The high risk of virus transmission, together with the existence of asymptomatic patients, forces dentists to consider all patients as potentially infected [5]. In addition, some health institutions worldwide, such as the National Health Commission (NHC) of the People’s Republic of China, the National Administration of Traditional Chinese Medicine, the Center for Disease Control and Prevention (CDC), The Association of Cantonal Dentists in Switzerland (VKZS) and the Swiss Dental Association (SSO) [9], the American Dental Association (ADA) [10], and the World Health Organization (WHO) [7], have recommended the use of oral antiseptics as agents that can reduce the viral load of SARS-CoV-2 before dental procedures [11].

Thus, the aim of the present study was to evaluate some of the antiseptic agents most frequently used by dental professionals and their antiviral effects against SARS-CoV-2 from both in vitro and in vivo studies to identify the most common compounds effective in reducing the viral load in the oral cavity in people infected with the virus.

## 2. Structure of SARS-CoV-2

Coronaviruses are a group of non-segmented positive-strand enveloped RNA viruses. SARS-CoV-2 particles have a spherical shape with diameters ranging from 60 to 140 nm and are coated with 9 to 12 nm spike proteins along with viral envelope proteins that give it the appearance of a crown [3]. The virus has a tropism for cells that have two specific receptors: angiotensin-converting enzyme 2 (ACE2) [12] and transmembrane serine protease-2 (TMPRSS2), which are mainly present in the lungs [13]. These receptors have been identified in epithelial cells of the tongue (spinous cell layer, stratum corneum, and epithelial surface), gingival tissues, oral mucosa, taste cells of human fungiform papillae, and salivary glands (ductal, acinar, and myoepithelial cells); thus, the oral cavity has gained relevance in the pathogenesis of the disease [13,14,15].

## 3. Importance of the Transmission and Pathogenicity of SARS-CoV-2 in the Oral Cavity

The highest SARS-CoV-2 viral loads are found in the nasal cavity, nasopharynx, and oropharynx in relation to the high expression of the ACE2 receptor [15]. The mouth is part of the oropharynx, and harbors bacteria and viruses from the nose, throat, respiratory tract, and contaminated saliva, which can spread viral infections [16].

In the first phase of COVID-19 infection (first 10 days), the patient is usually asymptomatic but highly contagious, and the virus accumulates mainly in the nasal, oral, and pharyngeal areas. The number of ACE2 receptors is greater in the salivary glands and in the mucosa of the tongue compared with the lungs and saliva droplets, which is why they represent the most relevant route of transmission [17]. The presence of the virus has been detected in 91.7% of saliva samples from patients with COVID-19, highlighting the importance of the oral cavity in the transmission and reservoir of SARS-CoV-2 [18].

Infection control in dental practice during the COVID-19 pandemic represents a challenge, because SARS-CoV-2 can be transmitted through aerosols produced by breathing, speaking, coughing, and sneezing [19]. Additionally, keeping a certain distance from patients increases the chance that any dental procedure will transmit the virus [19]. Dental procedures generate bioaerosols due to the use of certain specialized instruments; thus, dentists are highly exposed to aerosols carrying microorganisms that can cause infections, such as SARS-CoV-2, found in the saliva of infected people [5].

## 4. Saliva as a Vector

There are two main routes of transmission of SARS-CoV-2 [2]. The first is indirect, where a person can release droplets larger than 5 μm in size, either by breathing, speaking, sneezing, or coughing. The virus does not stay in the air but stays on surfaces, resulting in indirect transmission through contact with objects. The second is directly from person to person, through droplet nuclei smaller than 5 μm that remain suspended in the air for significant periods of time and enable transmission over distances greater than 1 m [17].

In dental procedures, the transmission of SARS-CoV-2 can easily occur during direct contact with patients, either through the inhalation of suspended airborne pathogens or through direct contact with biological fluids such as blood and saliva, and through surfaces or contaminated instruments. Studies have found that saliva droplets >60 μm transmit the virus at distances between 1 and 3 m, unlike saliva droplets <60 μm, where transmission reaches distances of up to 7–8 m. Coughing once or having a five-minute conversation can produce up to 3000 droplets of saliva, whereas a sneeze can produce up to 40,000 droplets [18]. The SARS-CoV-2 viral load in saliva is highest during the first week after symptom onset, and many patients may remain asymptomatic for a prolonged period, highlighting the importance of developing a strategy to prevent spread of the virus in the general population [3].

## 5. Recommendation of the Use of Oral Antiseptics

Some health and dental institutions worldwide have established the use of oral antiseptics that attack the lipid envelope of SARS-CoV-2 and can reduce the viral load in the mouth, nasopharynx, and oropharynx as a recommendation prior to any procedure performed in the mouth (dental exploration or treatment) [19]. SARS-CoV-2 is vulnerable to oxidation; therefore, a pre-procedure mouthwash containing oxidizing agents such as 1% hydrogen peroxide or 0.2% povidone–iodine is recommended to reduce the microbial load in saliva and the possible transport of SARS-CoV-2. A pre-procedure mouthwash would be most helpful in cases where a rubber dam cannot be used [20]. The oral viral load of SARS-CoV-2 has been associated with the severity of COVID-19; thus, its reduction could be associated with a decrease in the severity of the condition and the risk of transmission [17]. Several in vitro studies and clinical trials have been conducted, using different oral antiseptic agents at different concentrations and exposure times, which can be implemented as a form of intervention to reduce the viral load of SARS-CoV-2 in saliva (Table 1 and Table 2).

## 6. Types of Oral Antiseptics against SARS-CoV-2

### 6.1. Hydrogen Peroxide (H_2_O_2_)

Hydrogen peroxide is a widely used chemical compound with antimicrobial properties, and its efficacy has been demonstrated on different human viruses, including influenza and coronavirus viruses. It mainly affects viruses with a lipid envelope, such as SARS-CoV-2, through the generation of oxygen free radicals (Figure 1) [18]. Some health institutions worldwide have suggested the use of disinfection methods to eliminate SARS-CoV-2 based on H_2_O_2._ Cold atmospheric plasma technology is a method that generates a high concentration of H_2_O_2_ that causes oxidation of amino acids, nucleic acids, and induces peroxidation of unsaturated fatty acid through interaction with membrane lipids, altering the function of membranes in microorganisms. Chen et al. found that cold at-mospheric plasma eliminated SARS-CoV-2 on the surface of living organisms within 180 s [37]. In addition to its potential benefits in infection control and wound healing, cold atmospheric plasma has also been applied in studies targeting hemostasis control, treatment of skin diseases, immunotherapy, and regenerative medicine [37]. On the other hand, one of the first strategies implemented to prevent SARS-CoV-2 transmission during dental procedures was the use of a preoperative rinse consisting of 1% hydrogen peroxide solution to reduce viral load, due to the vulnerability of the virus to oxidation [3]. In vitro studies concluded that clinically recommended and commercially available concentrations of 1.5% and 3.0% H_2_O_2_ rinses showed minimal virucidal activity at 15 s and nearly the same effect at 30 s after application [19] (Table 1). Other in vitro studies compared the virucidal effect of a mouthwash with a concentration of 1.5% H_2_O_2_ versus mouthwashes with other active ingredients (povidone–iodine [PVP-I], chlorhexidine [CHX], ethanol+essential oils) at 30 s of exposure, showing that the virucidal activity of H_2_O_2_ was less effective [21,38]. However, Xu et al. reported a >99.9% clearance rate of SARS-CoV-2 after a contact time of 30 min [2] (Table 1). Carrouel et al., recommended three daily mouthwashes and two nasal washes from the onset of symptoms of COVID-19, during its evolution and in the hospitalization of cases without complications [18]. This measure, interestingly, could be applied to hospitalized patients, because the risk of transmission is increased in procedures such as ventilation, intubation, and non-invasive aspiration, which generate bioaerosols and result in nosocomial transmission from hospitalized infected people to healthy family members, caregivers, health professionals, and other patients in the hospital [39]. In a randomized controlled clinical trial, Chaudhary et al. showed that a mouth rinse with 1% H_2_O_2_ achieved mean viral load reductions of 61% to 89% at 15 min after application, and 70% to 97% at 45 min [33] (Table 2). In this sense, Eduardo et al. demonstrated that a 1.5% H_2_O_2_ rinse maintained a significant decrease in SARS-CoV-2 viral load up to 30 min after application, as compared with mouthwashes containing 0.075% cetylpyridinium chloride (CPC) + 0.28% zinc lactate and 0.12% chlorhexidine, which maintained a significant decrease in viral load up to 60 min after use [34] (Table 2). Meyers et al., evaluated various nasal and oral rinses in vitro, and tested three mouthwashes that included 1.5% H_2_O_2_ as the active ingredient which showed similar abilities to inactivate SARS-CoV-2, with reductions of viral load of <90% to 99% at 30 s during contact time and >90% to <99.99% at 1 min of exposure [27] (Table 1). However, not all studies have shown a reduction in viral load after using H_2_O_2_ rinses. In their clinical study, Gottsauner et al., concluded that a mouth rinse with 1% H_2_O_2_ does not reduce the intraoral viral load in subjects positive for SARS-CoV-2 (Table 2); therefore, additional studies are needed to evaluate the use of mouth rinses containing other active agents to reduce the intraoral load of SARS-CoV-2 [3]. Mouthwashes with a low concentration of H_2_O_2_ (≤ 1.5%) are safe, even when used over a long-term period. However, mouthwashes with a concentration of 3% H_2_O_2_ have shown adverse effects when used for 1 to 2 min [40]. These undesirable effects include a mouth pain sensation, burning sensation, erythema, edema, ulcers, and/or erosive changes in the oral mucosa [41].

### 6.2. Chlorhexidine (CHX)

Chlorhexidine is an antiseptic and biguanide disinfectant, with widely demonstrated antimicrobial activity against Gram-positive, Gram-negative, anaerobic, and aerobic bacteria, as well as some viruses and yeasts [17]. The mechanism of action of the biguanides is based on the strong association of the biguanide group with the anions exposed in the membrane and cell wall of the microorganism, particularly acid phospholipids and proteins (Figure 1) [42]. This is considered the gold standard for biofilm control, and its side effects are well known. CHX can be effective against enveloped viruses (herpes simplex virus, HIV, influenza virus, cytomegalovirus), but not against small non-enveloped viruses (enterovirus, poliovirus, human papillomavirus) [17]; in this context, it could also be effective against SARS-CoV-2 [5]. One study evaluated its effectiveness against SARS-CoV-2 at concentrations of 0.2% and 0.12% (Table 1). Its viricidal effect at 30 s was lower than povidone–iodine (PVP-I) [23,38]. Jain et al. found that its viricidal effect was the same at 30 and 60 s of exposure, although it was slightly higher if used at a 0.2% concentration [23] (Table 1). Xu et al., carried out an in vitro study where they observed that the potency of CHX was greater when the product was not washed off after virus binding, suggesting that the prolonged effect of rinses on cells affects antiviral outcome [2] (Table 1). Yoon et al., evaluated the effect on the salivary viral load after gargling at 1, 2, and 4 h with a CHX rinse; viral load was low at 2 h after gargling, but then increased [30] (Table 2). A randomized controlled clinical trial by Chaudhary et al. showed that a CHX rinse at 0.12% over 60 s achieved a mean reduction of 61% to 89% of viral load after 15 min, and a reduction of 70% to 97% after 45 min [33] (Table 2). Huang et al. concluded that in patients who combined the use of a mouth rinse and oropharyngeal spray with 0.12% CHX, they eliminated SARS-CoV-2 in 86% compared with 6.3% of a control group [35] (Table 2). The most commonly observed side effects associated with the long-term use of 0.06%, 0.12%, and 0.2% concentrations of CHX include a loss of taste, numbness, burning sensation, oral mucosal pain, (including tongue and gums), dryness, hypersensitivity reactions, or photosensitivity [43,44,45,46]. Heidari et al., determined that loss of taste is significantly greater in 0.2% CHX compared with concentrations at 0.12% and 0.06%, from the seventh day of use [45]. In addition, extrinsic brown stains may appear on teeth, dentures, composite restorations, and the tongue. The severity of dental stains can vary between patients and worsen when people also consume tea, port, red wine, and other tannin-containing substances [46].

### 6.3. Povidone–Iodine (PVP-I)

Povidone–iodine is an oxidizing agent composed of iodine and the water-soluble polymer polyvinylpyrrolidone. It exerts antimicrobial activity when it dissociates and releases iodine, which penetrates and alters protein synthesis, oxidizes nucleic acids, and lyses bacteria, fungi, and viruses (Figure 1) [18]. The recommendation of using an oral antiseptic with PVP-I in patients with COVID-19 is based on its virucidal activity against enveloped and non-enveloped viruses, including Ebola, Middle East respiratory syndrome (MERS), coronavirus (SARS), influenza, and hand, foot, and mouth disease (Coxsackievirus) [17].

The different commercial presentations of PVP-I, which include antiseptic solution (10%), hand sanitizer (7.5%), throat spray (0.45%), and mouthwash (1%), show reductions in the viral load of SARS-CoV-2, with a clearance rate of >99.99% after an exposure time of 30 s [22] (Table 1). Meyers et al. evaluated various nasal and oral rinses in vitro and concluded that a 5% PVP-I solution inactivates SARS-CoV-2 by >90% to <99.9% in a minimum contact time of 30 s, unlike three other mouth rinses that contain 1.5% H_2_O_2_, which reached the same inactivation at 1 min of contact time [27] (Table 1). Pelletier et al. performed an in vitro study with three nasal and three oral antiseptics with different concentrations of PVP-I; they concluded that all presentations tested were effective in inactivating SARS-CoV-2 after 60 s of exposure [28] (Table 1). Another study concluded that PVP-I at concentrations of 0.5%, 1%, and 1.5% completely inactivated SARS-CoV-2 within 15 s of contact [16] (Table 1). In comparison with H_2_O_2_, PVP-I exhibited a better virucidal activity at 15 s of exposure [19] (up to a threefold greater effect) (Table 1). Regarding clinical trials, Chaudhary et al. showed that a 0.5% PVP-I rinse for 60 s achieved a mean reduction of 61% to 89% in the viral load after 15 min, and a reduction of 70% to 97% after 45 min [33] (Table 2). Elzein et al. conducted a randomized controlled clinical trial which demonstrated that a mouth rinse with 1% PVP-I significantly reduced the intraoral viral load in subjects positive for SARS-CoV-2 [36] (Table 2). Khan et al. recommended the application of 0.5% PVP-I nasal drops in addition to 0.5% PVP-I mouth rinses for 30 s to achieve a reduction in SARS-CoV-2 viral load. However, the period of time during which antisepsis remains needs to be investigated through several randomized controlled studies [31] (Table 2).

The cytotoxic effects and tolerance of PVP-I are important points to consider for its implementation, because it is toxic to the oral and nasal mucosa at concentrations above 2.5% and 5%, respectively, although commercial formulations do not reach these concentrations [17]. Furthermore, it can be safely administered for five months in the nasal cavity and six months in the oral cavity. Prolonged use of PVP-I in concentrations of 1% to 1.25% does not irritate the mucous membranes or produce adverse effects for up to 28 months, nor does it stain the teeth or alter taste functions [47,48]. Its implementation has not been shown to affect thyroid function, but an increase in serum thyroid-stimulating hormone has been observed in individuals undergoing prolonged PVP-I treatment (24 weeks) [39]. Contraindications should be considered in patients with an anaphylactic allergy to iodine, pregnancy, active thyroid disease, and patients receiving radioactive iodine therapy. The alternative to PVP-I oral solution is the use of a 1.5% H_2_O_2_ rinse, as recommended by interim ADA guidelines [16].

### 6.4. Cetylpyridinium Chloride (CPC)

Cetylpyridinium chloride is a water-soluble, non-oxidizing, and non-corrosive quaternary ammonium compound which is highly cationic at neutral pH. In vitro studies have shown that it is capable of eliminating or inactivating different strains of the influenza virus (AH3N2, AH1N1) [17]. The antiviral mechanism of action of CPC lies in its ability to break the lipid envelope, interfering with the ability of the virus to enter the cell [17].

Li et al., suggest that a CPC concentration between 0.05% and 0.010% can reduce the viral load of SARS-CoV-2 [5,16]. Additionally, Koch-Heier et al., evaluated two rinses—ViruProX® (0.05% CPC and 1.5% H_2_O_2_) and BacterX® (0.1% CHX, 0.05% CPC, and 0.005% fluoride)—and their results showed that 0.1% CHX and 1.5% H_2_O_2_ present in the rinses did not exhibit a reduction in viral load, although 0.05% CPC present in both rinses was responsible for the virucidal effect against SARS-CoV-2, which significantly reduced the viral load [24] (Table 1). Tiong et al. found that CPC was able to decrease around 99.99% of the concentration of SARS-CoV-2 after 30 s of exposure, exhibiting a slightly higher virucidal activity than CHX [26] (Table 1).

In their clinical trials, Seneviratne et al. concluded that a CPC-containing mouth rinse decreased salivary viral load at 5 min of use in patients with COVID-19. In addition, they observed that the effect was maintained up to 3 and 6 h after application compared with a control group [32] (Table 2). Although CPC has demonstrated antiviral activity against several viruses that cause respiratory infections, more research is needed to elucidate the action of CPC against SARS-CoV-2 [18].

### 6.5. Bioflavonoids

Bioflavonoids and hydroxylated phenolic structures produced by plants have demonstrated their ability to eliminate bacteria, fungi, and viruses [49].

Hesperidin is a bioflavonoid contained in citrus peel and can interact with the SARS-CoV-2 glycoproteins responsible for cell infection and virus replication; thus, it is likely that it inhibits binding of the virus to its ACE2 receptor. Other flavonoids, such as naringin, caflanone, equivir (a mixture of bioflavonoids), hesperetin, and myricetin, may exert a similar action to hesperidin [18].

### 6.6. Ethanol

Ethanol can dissolve the lipid membrane and denaturing proteins from a variety of microorganisms and is active in high concentrations for the inactivation of enveloped viruses. It is available in different concentrations in many oral antiseptics with formulations graduated from 14% to 27% [18]. In an in vitro study, a solution of 70% ethanol was unable to completely inactivate SARS-CoV-2 after 15 s of contact, but was effective at 30 s of exposure [16] (Table 1).

### 6.7. Essential Oils

Some studies have suggested the benefits of essential oils in controlling viral contamination, at least for herpes viruses [17]. Davies et al. evaluated several commercially available mouthwashes, including Listerine Total Care^®^, and showed that it significantly reduced SARS-CoV-2 titers after rinsing for 1 min. However, the contribution of its active ingredients (eucalyptol, thymol, menthol, sodium fluoride, and zinc fluoride) to its antiviral activity is not clear [25] (Table 1). The most commonly used active ingredients are eucalyptol at 0.092%, thymol at 0.064%, methyl salicylate at 0.060%, and menthol at 0.042% [5].

Limonene is a monoterpene found in natural fruits such as grapefruit (95%), tangerine (94%), orange (91%), lemon (65%), and elemi (50%) [49,50]. It is frequently used as a dietary supplement and as a fragrance ingredient for cosmetic products because it is considered safe [49,51]. Rodríguez-Casanovas et al., evaluated the virucidal activity of different mouthwashes and found that D-limonene exhibited a significant reduction in virucidal activity of around 99.99% against SARS-CoV-2, through a solution containing D-limonene (0.2%) and CPC (0.05%) [29].

## 7. Real Virucidal Effect or Cytotoxic Effect of the Rinse?

Possible inhibitory effects of oral antiseptics on SARS-CoV-2 inactivation have been proposed based on the assumption that organic components of oral antiseptics disrupt viral envelopes or act on viral proteins. Viable cells are required for productive infection; therefore, the toxic effects on cells can be misinterpreted as a potent antiviral effect. Although most commercially available oral antiseptics are safe, their effects on virus infectivity may be overestimated if mouth-rinse-associated cytotoxicity is not considered, because antiseptic-associated cell death may result in a decrease in the number of target cells for viral infection, producing an apparently decreased viral infectivity. In their in vitro study, Xu et al., concluded that all oral antiseptics at non-cytotoxic levels exhibited antiviral activity; however, their cytotoxic effects must be taken into account when evaluating antiviral activity [2].

## 8. Discussion

Dentists experience the highest risk of exposure to COVID-19 infection due to the dental devices and instruments used (ultrasound, handpieces, triple syringes, etc.) that can generate large amounts of aerosols, which disperse numerous bacteria and viruses [5]. The presence of SARS-CoV-2 has been identified in different types of materials, remaining in plastic for up to 4 days, and 7 days in stainless steel and surgical masks [8]; for this reason, additional measures are being sought that could reduce the viral load of SARS-CoV-2 and thereby reduce its transmission. Oral rinses could significantly reduce the viral load because saliva is one of the main vectors. Coughing once or having a five-minute conversation can produce up to 3000 droplets of saliva, whereas a sneeze can produce up to 40,000 droplets [18]. It has recently been shown that mouthwashes can rapidly inactivate SARS-CoV-2 through in vitro [2,16,19,21,22,23,24,25,26,27,28,29] and in vivo studies [3,30,31,32,33,34,35,36]. Subsequently, there have been extensive discussions regarding the utilization of mouth rinses to possibly complement current prevention measures such as facemasks, hand disinfection, and social distancing in order to reduce the global spread of SARS-CoV-2 [18]. To identify the effect of oral antiseptics, we conducted a literature review of the active ingredients that have demonstrated effects on SARS-CoV-2 both in vitro and in vivo. The efficient inactivation of coronaviruses (SARS and MERS) on inanimate surfaces using hydrogen peroxide (0.5% H_2_O_2_ for 1 min) was evaluated by Kampf et al., demonstrating good results [52]. H_2_O_2_ application was a one of the first strategies implemented to prevent SARS-CoV-2 transmission during dental procedures is the use of a preoperative rinse consisting of 1% hydrogen peroxide solution to reduce viral load, due to the vulnerability of the virus to oxidation [3]. Meyers et al. evaluated various nasal and oral rinses in vitro, and tested three mouthwashes that included 1.5% H_2_O_2_ as the active ingredient that showed similar abilities to inactivate SARS-CoV-2 with a reduction in viral load of <90% to 99% at 30 s during a contact time and >90% to <99.99% at 1 min of exposure [27]. For this reason, it has been proposed that hydrogen peroxide, as an antiseptic agent, could play a fundamental role in reducing the rate of hospitalization and complications associated with COVID-19. The antiseptic action is due not only to the known oxidative and mechanical scavenging properties of hydrogen peroxide, but also to the induction of the innate antiviral inflammatory response through overexpression of Toll-like receptor 3 (TLR3) [53]. No damage to the oral mucous membranes or their microvilli was observed after continuous treatment with 3% H_2_O_2_ gargling [54].

CHX is considered to be the gold standard for biofilm control; it can be effective against enveloped viruses (such as herpes simplex virus, HIV, influenza virus, and cytomegalovirus), but not against small non-enveloped viruses (such as enterovirus, poliovirus, and human papillomavirus) [17]; in this context, it could also be effective against SARS-CoV-2 [5]. A high level of virus in saliva was detected in a clinical trial performed by Yoon et al. in 2020, but CHX was able to significantly decrease the viral load for 2 h after a single use [30].

Huang et al. concluded that in patients who combined the use of a mouth rinse and oropharyngeal spray with 0.12% CHX, they eliminated SARS-CoV-2 in 86% of subjects compared with 6.3% of a control group [35]. In summary, it is concluded that CHX may exert an interesting virucidal efficacy against HSV-1 and influenza A viruses. However, reductions in SARS-CoV-2 strains have not yet been demonstrated when evaluated in vitro. The use of a CHX mouthwash was identified to temporarily reduce the viral load of SARS-CoV-2 in patients with COVID-19 [30,35].

Regarding PVP-I, Meyers et al. evaluated various nasal and oral rinses in vitro and concluded that a 5% PVP-I solution inactivates SARS-CoV-2 by >90% to <99.9% after a minimum contact time of 30 s, unlike three other mouth rinses containing 1.5% H_2_O_2_, which reached the same inactivation at 1 min of contact time [27]. Elzein et al., conducted a randomized controlled clinical trial which demonstrated that a mouth rinse with 1% PVP-I significantly reduced the intraoral viral load in subjects positive for SARS-CoV-2 [36]. However, the period of time during which antisepsis remains needs to be investigated through several randomized controlled studies [31]. Komine et al., also showed that mouth rinses containing 0.04–0.075% CPC inactivated >99.99% of SARS-CoV-2 in 20–30 s [55].

Koch-Heier et al. evaluated two rinses—0.05% CPC and 1.5% H_2_O_2_, and 0.1% CHX, 0.05% CPC, and 0.005% fluoride—and their results showed that rinses with 0.1% CHX and 1.5% H_2_O_2_ did not result in a reduction in viral load, but that 0.05% CPC present in both rinses was responsible for the virucidal effect against SARS-CoV-2, which significantly reduced the viral load [24].

Although CPC has demonstrated antiviral activity against several viruses that cause respiratory infections, more research is needed to elucidate the action of CPC against SARS-CoV-2 [18]. Rodríguez-Casanovas et al. evaluated the virucidal activity of different mouthwashes, and found that D-limonene resulted in a significant reduction in virucidal activity of around 99.99% against SARS-CoV-2, through a solution containing D-limonene (0.2%) and CPC (0.05%) [29].

## 9. Conclusions

The oral cavity is an important source of SARS-CoV-2 transmission and plays an important role in the pathogenesis of COVID-19. Dentists are exposed to contamination during dental care; thus, it is necessary to prevent transmission by implementing cleaning and disinfection strategies and avoiding elective dental procedures that favor the generation of aerosols. The most commonly recommended oral antiseptics against SARS-CoV-2 are PVP-I, H_2_O_2_, CHX, and D-limonene, combined with CPC. There is sufficient in vitro evidence to support the use of antiseptics, either for dental examination or treatment, to potentially reduce the viral load of SARS-CoV-2 or other coronaviruses. Adopting these simple preventive measures, oral health professionals can reduce their risk of infection by reducing the viral load in the mouth of patients. However, it is important to highlight that the impact of a “false sense of security” on health professionals and patients and society should not be underestimated, because this can lead to a reduction in the use of protective equipment or closer social interactions with potentially infected people, thereby increasing SARS-CoV-2 infections. Currently, clinical trials and in vivo studies on the virucidal effect against SARS-CoV-2 are limited; thus, more research of this type is needed in order to implement protocols or clinical practice guidelines against SARS-CoV-2 or other emerging microorganisms. In addition, it is necessary for health professionals to be up to date in infection control, even more so in pandemics such as COVID-19, where the use of antiseptics could be enormous help in preventing the spread of SARS-CoV-2.

## Figures and Tables

**Figure 1 ijerph-19-08768-f001:**
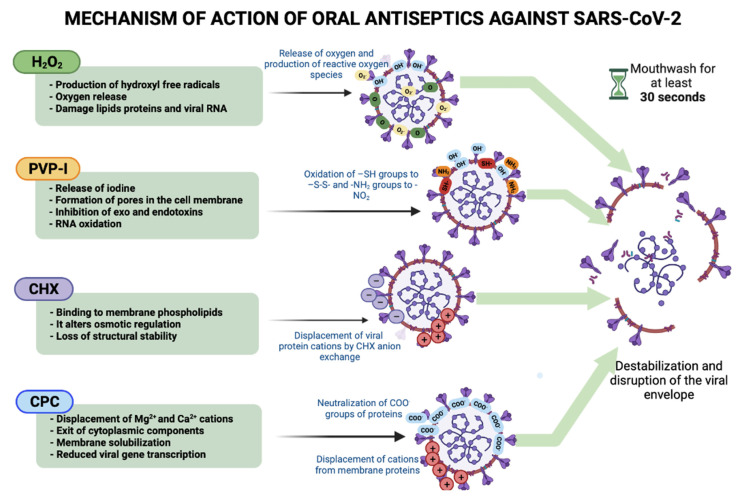
Mechanism of oral antiseptics. Hydrogen peroxide (H_2_O_2_) produces hydroxyl free radicals and reactive oxygen species that react with lipids, proteins, and RNA. Povidone–iodine (PVP-I) oxides –SH groups to –S–S– and –NH_2_ groups to –NO_2_. Chlorhexidine (CHX) binds to membrane phospholipids and displaces viral protein cations by CHX anion exchange. Cetylpyridinium chloride (CPC) displaces cations and neutralizes negative –COO– charges of proteins, which breaks the viral membrane. Figure created with BioRender, © biorender.com.

**Table 1 ijerph-19-08768-t001:** In vitro assays of the antiviral efficacy of active ingredients against SARS-CoV-2.

Active Ingredients	Effect	Year	References
Povidone–iodine (PVP-I); hydrogen peroxide (H_2_O_2_)	After 15 and 30 s of contact, PVP-I at 0.5%, 1.25%, and 1.5% completely inactivated SARS-CoV-2. The solutions of H_2_O_2_ at 1.5% and 3.0% showed minimal virucidal activity.	2020	[19]
Hydrogen peroxide (H_2_O_2_); chlorhexidine digluconate; a solution with benzalkonium chloride, dequalinium chloride; 1.0% PVP-I solution; a mouthwash with ethanol and essential oils; a solution with octenidine dihydrochloride; polyaminopropylbiguanide solution.	All the active ingredients inactivated different strains of SARS-CoV-2. Particularly, the solution with benzalkonium chloride, dequalinium chloride, PVP-I solution, and mouthwash with ethanol and essential oils significantly reduced viral infectivity to undetectable levels.	2020	[21]
PVP-I 10% *w*/*v* solution; PVP-I 7.5% *w*/*v* solution; PVP-I 1.0% *w*/*v* solution and PVP-I 0.45% *w*/*v* solution.	All the PVP-I solutions inactivated SARS-CoV-2 by 99.99% at 30 s of exposure.	2020	[22]
PVP-I oral antiseptic solutions at concentrations of 0.5%, 1.0%, and 1.5%; ethanol at 70%.	The three concentrations of povidone–iodine completely inactivated SARS-CoV-2 in the first 15 s of contact. Ethanol at 70% inactivated the virus at 30 s of exposure.	2020	[16]
Chlorhexidine digluconate, 0.2%	Within 30 s of exposure, it inactivated 99.99% of SARS-CoV-2 and was even more effective than povidone–iodine.	2021	[23]
H_2_O_2_ solution; (alcohol and essential oils); povidone–iodine and chlorhexidine gluconate.	A 5% *v*/*v* dilution of H_2_O_2_ or povidone-iodine completely inhibited SARS-CoV-2 infectivity; a 50% *v*/*v* dilution of chlorhexidine gluconate or (alcohol and essential oils) was necessary to inhibit the capacity of SARS-CoV-2 infection.	2021	[2]
Oral antiseptic solution with 0.05% cetylpyridinium chloride (CPC) and 1.5% H_2_O_2_; oral antiseptic solution with 0.1% chlorhexidine, 0.05% CPC and 0.005% sodium fluoride.	CPC or a combination of chlorhexidine with CPC is more effective than chlorhexidine or H_2_O_2_ alone. Rinses with these active ingredients could reduce the viral load in the oral cavity and the transmission of SARS-CoV-2 during dental procedures.	2021	[24]
A solution with 1.4% dipotassium oxalate, without ethanol; mouthwash with eucalyptol, thymol, menthol, sodium fluoride, and zinc fluoride.	These active ingredients, as well as other mouthwashes containing 0.01–0.02% hypochlorous acid or 0.58% povidone–iodine, successfully inactivated SARS-CoV-2.	2021	[25]
0.12% chlorhexidine digluconate solution; antiseptic oral solution with 0.075% CPC and 0.05% NaF; 0.05% thymol solution; mouthwash with 0.1% hexetidine with 9% ethanol; saline water (0.34 M sodium chloride).	Rinses with CPC and hexetidine demonstrated a potent virucidal effect. Chlorhexidine digluconate showed a slightly lesser effect, whereas thymol or saline water had no significant effect in reducing the SARS-CoV-2 viral load.	2021	[26]
Sodium bicarbonate (NaHCO_3_; 1% baby shampoo in PBS (phosphate-buffered saline) nasal rinse solution; 1.5% H_2_O_2_ solution; oral antiseptic solution with 1.5% H_2_O_2_ and 0.1% menthol 0.1%; 0.07% CPC solution; mouthwash solutions with 0.092% eucalyptol, 0.042% menthol, 0.06% methyl salicylate, and 0.064% thymol; 5% PVP-I solution.	1% baby shampoo nasal rinse solution and mouthwash solutions with eucalyptol, menthol, methyl salicylate, and thymol inactivated 99.9% of the human coronavirus. Rinses with 1.5% and 3.0% H_2_O_2_ decreased the viral load of human coronaviruses between 90% and 99%. PVP-I exhibited a virucidal effect on human coronaviruses, but NaHCO_3_ had no effect on coronavirus viral load.	2021	[27]
Oral and nasal antiseptic solutions with 1% or 5% PVP-I.	All concentrations of PVP-I showed a virucidal effect against SARS-CoV-2.	2021	[28]
Solution with D-limonene (0.2%) and CPC (0.05%).	It showed an approximately 99.99% reduction in virucidal activity against SARS-CoV-2.	2022	[29]

**Table 2 ijerph-19-08768-t002:** Clinical trials and in vivo assays of the antiviral efficacy of active ingredients against SARS-CoV-2.

Active Ingredients	Effect	Year	References
1% H_2_O_2_ solution.	Mouthwash with 1% H_2_O_2_ did not produce a significant reduction in intraoral viral load in patients positive for SARS-CoV-2 after 30 min of application.	2020	[3]
0.12% chlorhexidine gluconate solution.	0.12% chlorhexidine solution considerably decreased the viral load of SARS-CoV-2 in the saliva of patients during early stages of COVID-19 infection.	2020	[30]
0.5% PVP-I solution.	The solution applied intranasally and orally was well tolerated in a group of patients and health workers. Pre-trial studies suggest that the use of 0.5% PVP-I could help in reducing the spread of SARS-CoV-2, especially during clinical examination procedures such as endoscopy.	2020	[31]
0.5% *w*/*v* PVP-I solution; 0.2% *w*/*v* chlorhexidine gluconate solution; 0.075% CPC solution.	PVP-I and CPC mouthwashes decreased the SARS-CoV-2 viral load in patient saliva samples up to 6 h after use.	2021	[32]
1% H_2_O_2_ solution; 0.12% chlorhexidine gluconate solution; 0.5% PVP-I solution; saline solution.	The four solutions decreased the viral load of SARS-CoV-2 in saliva samples from asymptomatic, pre-symptomatic, symptomatic, and post-symptomatic individuals. In those individuals with an initial viral load of fewer than 104 copies/mL of saliva, there was a 100% reduction in viral load at 15 and 45 min after application.	2021	[33]
Mouthwash with 0.075% CPC + 0.28% zinc lactate; mouthwash with 1.5% H_2_O_2_; mouthwash with 0.12% chlorhexidine gluconate.	Mouthwashes with CPC + zinc lactate and chlorhexidine gluconate significantly decreased the viral load of SARS-CoV-2 in the saliva of patients up to 60 min after application, whereas H_2_O_2_ only decreased viral load 30 min after use.	2021	[34]
0.12% chlorhexidine gluconate.	The use of chlorhexidine gluconate as a mouthwash applied for 30 s twice a day for 4 days was effective in eliminating the viral load of SARS-CoV-2 in 62.1% of a group of 121 patients positive for infection. Additionally, 173 patients positive for SARS-CoV-2 and treated with chlorhexidine gluconate mouthwash and nasal spray twice daily for 4 days resulted in clearance of the SARS-CoV-2 viral load in 86% of patients.	2021	[35]
0.2% chlorhexidine gluconate; 1% PVP-I.	The application of oral solutions with chlorhexidine and PVP-I for 30 s was effective in significantly reducing the viral load of SARS-CoV-2 in the saliva of 61 patients positive for infection.	2021	[36]

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
