# Peer review of "Oral Antiseptics against SARS-CoV-2: A Literature Review"

_ijerph, 2022, doi:10.3390/ijerph19148768_

Round 1

Reviewer 1 Report

The manuscript (ijerph-1770627) entitled: Oral Antiseptics against SARS-CoV-2: A Literature Review. In this review article, the authors report a series of oral antiseptics which have activity against SARS-COV-2 and they also analyze the different options of oral antiseptics used in dental infection control. This work has been carried out correctly and it is in the fields of the journal.  The manuscript can be accepted after addressing the following issues.

1.      In the opinion of this reviewer, No need to mention Commercial mouthwashes, only the name of the active ingredients is enough.

2.      It might be helpful to strengthen the references that support the hypothesis that patients who use oral antiseptic have fewer complications when infected with the virus.

3.      Both tables 1 and 2 are confusing, please reorganize.

4.      Author may add some recent references of the field

5.      The reference should be carefully checked. The formatting is not homogenous.

6.      There are additional numerous places throughout the manuscript where relatively minor edits in sentence structure and/or grammar would benefit the overall manuscript. This reviewer will not attempt to address all of these beyond what has already been mentioned (comments above) but it should be addressed by the authors prior to publication.

Author Response

REVIEWER 1

  1. In the opinion of this reviewer, no need to mention Commercial mouthwashes, only the name of the active ingredients is enough.

Trade names of mouthwashes were removed from the text and from tables 1 and 2.

  1. It might be helpful to strengthen the references that support the hypothesis that patients who use oral antiseptic have fewer complications when infected with the virus.

The primary objective of this research was to evaluate the decrease in viral load in the oral cavity and thereby reduce the professional risk due to contamination with aerosols. Our objective was not to identify the decrease in complications in infected patients. A discussion was added where the probable decrease in complications caused by the virus is discussed.

Discussion

Dentists experience the highest risk of exposure to COVID-19 infection due to the dental devices and instruments used (ultrasound, handpieces, triple syringes, etc.) that can generate large amounts of aerosols, which disperse numerous bacteria and viruses[5]. The presence of SARS-CoV-2 has been identified in different types of materials, remaining in plastic for up to 4 days, and 7 days in stainless steel and surgical masks[8]; for this reason, additional measures are being sought that could reduce the viral load of SARS-CoV-2 and thereby reduce its transmission. Oral rinses could significantly reduce the viral load because saliva is one of the main vectors. Coughing once or having a five-minute conversation can produce up to 3,000 droplets of saliva, whereas a sneeze can produce up to 40,000 droplets [18]. It has recently been shown that mouthwashes can rapidly inactivate SARS-CoV-2 through in vitro [2, 16, 19, 21-29] and in vivostudies [3, 30-36]. Subsequently, there have been extensive discussions regarding the utilization of mouth rinses to possibly complement current prevention measures such as facemasks, hand disinfection, and social distancing in order to reduce the global spread of SARS-CoV-2[18]. To identify the effect of oral antiseptics, we conducted a literature review of the active ingredients that have demonstrated effects on SARS-CoV-2 both in vitro and in vivo. The efficient inactivation of coronaviruses (SARS and MERS) on inanimate surfaces using hydrogen peroxide (0.5% H2O2 for 1 minute) was evaluated by Kampf et al., demonstrating good results [51]. H2O2 application was a one of the first strategies implemented to prevent SARS-CoV-2 transmission during dental procedures is the use of a preoperative rinse consisting of 1% hydrogen peroxide solution to reduce viral load, due to the vulnerability of the virus to oxidation[3]. Meyers et al. evaluated various nasal and oral rinses in vitro, and tested three mouthwashes that included 1.5% H2O2 as the active ingredient that showed similar abilities to inactivate SARS-CoV-2 with a reduction in viral load of <90% to 99% at 30 s during a contact time and >90% to <99.99% at 1 min of exposure[27]. For this reason, it has been proposed that hydrogen peroxide, as an antiseptic agent, could play a fundamental role in reducing the rate of hospitalization and complications associated with COVID-19. The antiseptic action is due not only to the known oxidative and mechanical scavenging properties of hydrogen peroxide, but also to the induction of the innate antiviral inflammatory response through overexpression of Toll-like receptor 3 (TLR3) [52]. No damage to the oral mucous membranes or their microvilli was observed after continuous treatment with 3% H2O2 gargling[53].

CHX is considered to be the gold standard for biofilm control; it can be effective against enveloped viruses (such as herpes simplex virus, HIV, influenza virus, and cytomegalovirus), but not against small non-enveloped viruses (such as enterovirus, poliovirus, and human papillomavirus)[17]; in this context, it could also be effective against SARS-CoV-2[5]. A high level of virus in saliva was detected in a clinical trial performed by Yoon et al. in 2020, but CHX was able to significantly decrease the viral load for 2 h after a single use [30].

Huang et al. concluded that in patients who combined the use of a mouth rinse and oropharyngeal spray with 0.12% CHX, they eliminated SARS‐CoV‐2 in 86% of subjects compared with 6.3% of a control group[35]. In summary, it is concluded that CHX may exert an interesting virucidal efficacy against HSV-1 and influenza A viruses. However, reductions in SARS-CoV-2 strains have not yet been demonstrated when evaluated in vitro. The use of a CHX mouthwash was identified to temporarily reduce the viral load of SARS-CoV-2 in patients with COVID-19 [30, 35].

Regarding PVP-I, Meyers et al. evaluated various nasal and oral rinses in vitro and concluded that a 5% PVP-I solution inactivates SARS-CoV-2 by >90% to <99.9% after a minimum contact time of 30 s, unlike three other mouth rinses containing 1.5% H2O2, which reached the same inactivation at 1 min of contact time[27]. Elzein et al. conducted a randomized controlled clinical trial which demonstrated that a mouth rinse with 1% PVP-I significantly reduced the intraoral viral load in subjects positive for SARS-CoV-2 [36]. However, the period of time during which antisepsis remains needs to be investigated through several randomized controlled studies [31].

Komine et al. also showed that mouth rinses containing 0.04–0.075% CPC inactivated >99.99% of SARS-CoV-2 in 20–30 s [54].

Koch-Heier et al. evaluated two rinses—0.05% CPC and 1.5% H2O2, and 0.1% CHX, 0.05% CPC, and 0.005% fluoride—and their results showed that rinses with 0.1% CHX and 1.5% H2O2 did not result in a reduction in viral load, but that 0.05% CPC present in both rinses was responsible for the virucidal effect against SARS-CoV-2, which significantly reduced the viral load [24].

Although CPC has demonstrated antiviral activity against several viruses that cause respiratory infections, more research is needed to elucidate the action of CPC against SARS-CoV-2[18]. Rodríguez-Casanovas et al. evaluated the virucidal activity of different mouthwashes, and found that D-limonene resulted in a significant reduction in virucidal activity of around 99.99% against SARS-CoV-2, through a solution containing D-limonene (0.2%) and CPC (0.05%)[29].

  1. Both tables 1 and 2 are confusing, please reorganize.

       Tables 1 and 2 were rearranged

Table 1. In vitro assays of the antiviral efficacy of active ingredients against SARS-CoV-2.

Active ingredients

Effect

Year

References

Povidone–iodine (PVP-I); hydrogen peroxide (H2O2)

After 15 and 30 s of contact, PVP-I at 0.5%, 1.25%, and 1.5% completely inactivated SARS-CoV-2. The solutions of H2O2 at 1.5% and 3.0% showed minimal virucidal activity.

2020

[19]

Hydrogen peroxide (H2O2); chlorhexidine digluconate; a solution with benzalkonium chloride, dequalinium chloride; 1.0% PVP-I solution; a mouthwash with ethanol and essential oils; a solution with octenidine dihydrochloride; polyaminopropylbiguanide solution.

All the active ingredients inactivated different strains of SARS-CoV-2. Particularly, the solution with benzalkonium chloride, dequalinium chloride, PVP-I solution, and mouthwash with ethanol and essential oils significantly reduced viral infectivity to undetectable levels.

2020

[21]

PVP-I 10% w/v solution; PVP-I 7.5% w/v solution; PVP-I 1.0% w/v solution and PVP-I 0.45% w/v solution.

All the PVP-I solutions inactivated SARS-CoV-2 by 99.99% at 30 s of exposure.

2020

[22]

PVP-I oral antiseptic solutions at concentrations of 0.5%, 1.0%, and 1.5%; ethanol at 70%.

The three concentrations of povidone–iodine completely inactivated SARS-CoV-2 in the first 15 s of contact. Ethanol at 70% inactivated the virus at 30 s of exposure.

2020

[16]

Chlorhexidine digluconate, 0.2%

Within 30 s of exposure, it inactivated 99.99% of SARS-CoV-2 and was even more effective than povidone–iodine.

2021

[23]

H2O2 solution; (alcohol and essential oils); povidone–iodine and chlorhexidine gluconate.

A 5% v/v dilution of H2O2 or povidone-iodine completely inhibited SARS-CoV-2 infectivity; a 50% v/v dilution of chlorhexidine gluconate or (alcohol and essential oils) was necessary to inhibit the capacity of SARS-CoV-2 infection.

2021

[2]

Oral antiseptic solution with 0.05% cetylpyridinium chloride (CPC) and 1.5% H2O2; oral antiseptic solution with 0.1% chlorhexidine, 0.05% CPC and 0.005% sodium fluoride.

CPC or a combination of chlorhexidine with CPC is more effective than chlorhexidine or H2O2alone. Rinses with these active ingredients could reduce the viral load in the oral cavity and the transmission of SARS-CoV-2 during dental procedures.

2021

[24]

A solution with 1.4% dipotassium oxalate, without ethanol; mouthwash with eucalyptol, thymol, menthol, sodium fluoride, and zinc fluoride.

These active ingredients, as well as other mouthwashes containing 0.01% - 0.02% hypochlorous acid or 0.58% povidone–iodine, successfully inactivated SARS-CoV-2.

2021

[25]

0.12% chlorhexidine digluconate solution; antiseptic oral solution with 0.075% CPC and 0.05% NaF; 0.05% thymol solution; mouthwash with 0.1% hexetidine with 9% ethanol; saline water (0.34 M sodium chloride).

Rinses with CPC and hexetidine demonstrated a potent virucidal effect. Chlorhexidine digluconate showed a slightly lesser effect, whereas thymol or saline water had no significant effect in reducing the SARS-CoV-2 viral load.

2021

[26]

Sodium bicarbonate (NaHCO3; 1% baby shampoo in PBS (phosphate-buffered saline) nasal rinse solution; 1.5% H2O2 solution; oral antiseptic solution with 1.5% H2O2 and 0.1% menthol 0.1%; 0.07% CPC solution; mouthwash solutions with 0.092% eucalyptol, 0.042% menthol, 0.06% methyl salicylate, and 0.064% thymol; 5% PVP-I solution.

1% baby shampoo nasal rinse solution and mouthwash solutions with eucalyptol, menthol, methyl salicylate, and thymol inactivated 99.9% of the human coronavirus. Rinses with 1.5% and 3.0% H2O2decreased the viral load of human coronaviruses between 90% and 99%. PVP-I exhibited a virucidal effect on human coronaviruses, but NaHCO3had no effect on coronavirus viral load.

2021

[27]

Oral and nasal antiseptic solutions with 1% or 5% PVP-I.

All concentrations of PVP-I showed a virucidal effect against SARS-CoV-2.

2021

[28]

Solution with D-limonene (0.2%) and CPC (0.05%).

It showed an approximately 99.99% reduction in virucidal activity against SARS-CoV-2.

2022

[29]

Table 2. Clinical trials and in vivo assays of the antiviral efficacy of active ingredients against SARS-CoV-2.

Active ingredients

Effect

  Year

References

1% H2O2 solution.

Mouthwash with 1% H2O2 did not produce a significant reduction in intraoral viral load in patients positive for SARS-CoV-2 after 30 min of application.

2020

[3]

0.12% chlorhexidine gluconate solution.

0.12% chlorhexidine solution considerably decreased the viral load of SARS-CoV-2 in the saliva of patients during early stages of COVID-19 infection.

2020

[30]

0.5% PVP-I solution.

The solution applied intranasally and orally was well tolerated in a group of patients and health workers. Pre-trial studies suggest that the use of 0.5% PVP-I could help in reducing the spread of SARS-CoV-2, especially during clinical examination procedures such as endoscopy.

2020

[31]

0.5% w/v PVP-I solution; 0.2% w/v chlorhexidine gluconate solution; 0.075% CPC solution.

PVP-I and CPC mouthwashes decreased the SARS-CoV-2 viral load in patient saliva samples up to 6 h after use.

2021

[32]

1% H2O2 solution; 0.12% chlorhexidine gluconate solution; 0.5% PVP-I solution; saline solution.

The four solutions decreased the viral load of SARS-CoV-2 in saliva samples from asymptomatic, pre-symptomatic, symptomatic, and post-symptomatic individuals. In those individuals with an initial viral load of fewer than 104 copies/mL of saliva, there was a 100% reduction in viral load at 15 and 45 min after application.

2021

 [33]

Mouthwash with 0.075% CPC + 0.28% zinc lactate; mouthwash with 1.5% H2O2; mouthwash with 0.12% chlorhexidine gluconate.

Mouthwashes with CPC + zinc lactate and chlorhexidine gluconate significantly decreased the viral load of SARS-CoV-2 in the saliva of patients up to 60 min after application, whereas H2O2only decreased viral load 30 min after use.

2021

   [34]

0.12% chlorhexidine gluconate.

The use of chlorhexidine gluconate as a mouthwash applied for 30 s twice a day for 4 days was effective in eliminating the viral load of SARS-CoV-2 in 62.1% of a group of 121 patients positive for infection. Additionally, 173 patients positive for SARS-CoV-2 and treated with chlorhexidine gluconate mouthwash and nasal spray twice daily for 4 days resulted in clearance of the SARS-CoV-2 viral load in 86% of patients.

2021

[35]

0.2% chlorhexidine gluconate; 1% PVP-I.

The application of oral solutions with chlorhexidine and PVP-I for 30 s was effective in significantly reducing the viral load of SARS-CoV-2 in the saliva of 61 patients positive for infection.

2021

   [36]

  1. Author may add some recent references of the field

Added more related references

  1. Organization, W. H. https://covid19.who.int/
  2. Xu, C.; Wang, A.; Hoskin, E. R.; Cugini, C.; Markowitz, K.; Chang, T. L.; Fine, D. H., Differential effects of antiseptic mouth rinses on SARS-CoV-2 infectivity in vitro. Pathogens 2021, 10, (272), 2-14.
  3. Gottsauner, M. J.; Michaelides, I.; Schmidt, B.; Scholz, K. J.; Buchalla, W.; Widbiller, M.; Hitzenbichler, F.; Ettl, T.; Reichert, T. E.; Bohr, C., A prospective clinical pilot study on the effects of a hydrogen peroxide mouthrinse on the intraoral viral load of SARS-CoV-2. Clinical oral investigations 2020, 24, (10), 3707-3713.
  4. de Seguridad, A.; Ocupacional, S., Guía sobre la Preparación de los Lugares de Trabajo para el virus, COVID-19. Recuperado de: https://www. osha. gov/Publications/OSHA3992. pdf 2020.
  5. Testori, T.; Wang, H.-L.; Basso, M.; Bordini, G.; Dian, A.; Vitelli, C.; Miletić, I.; Del Fabbro, M., COVID-19 and Oral Surgery: A narrative review of preoperative mouth rinses. Acta Stomatologica Croatica 2020, 54, (4), 431-441.
  6. Robertson, C.; Clarkson, J. E.; Aceves-Martins, M.; Ramsay, C. R.; Richards, D.; Colloc, T.; Group, C. W., A Review of Aerosol Generation Mitigation in International Dental Guidance. International dental journal 2021, 72, (2), 203-210.
  7. Organization, W. H., Considerations for the provision of essential oral health services in the context of COVID-19: interim guidance, 3 August 2020. In World Health Organization: 2020; pp 1-5.
  8. Chin, A. W.; Chu, J. T.; Perera, M. R.; Hui, K. P.; Yen, H.-L.; Chan, M. C.; Peiris, M.; Poon, L. L., Stability of SARS-CoV-2 in different environmental conditions. The Lancet Microbe 2020, 1, (1), e10.
  9. Jiang, C. M.; Duangthip, D.; Auychai, P.; Chiba, M.; Folayan, M. O.; Hamama, H. H. H.; Kamnoedboon, P.; Lyons, K.; Matangkasombut, O.; Mathu-Muju, K. R., Changes in oral health policies and guidelines during the COVID-19 pandemic. Frontiers in Oral Health 2021, 2, 1-14.
  10. Elmahgoub, F.; Coll, Y., Could certain mouthwashes reduce transmissibility of COVID-19? Evidence-Based Dentistry 2021, 22, (2), 82-83.
  11. Deana, N. F.; Seiffert, A.; Aravena-Rivas, Y.; Alonso-Coello, P.; Muñoz-Millán, P.; Espinoza-Espinoza, G.; Pineda, P.; Zaror, C., Recommendations for Safe Dental Care: A Systematic Review of Clinical Practice Guidelines in the First Year of the COVID-19 Pandemic. International journal of environmental research and public health 2021, 18, 2-19.
  12. Ortega, K.; Rech, B.; El Haje, G.; Gallo, C.; Pérez-Sayáns, M.; Braz-Silva, P., Do hydrogen peroxide mouthwashes have a virucidal effect? A systematic review. Journal of Hospital Infection 2020, 106, (4), 657-662.
  13. Zhu, F.; Zhong, Y.; Ji, H.; Ge, R.; Guo, L.; Song, H.; Wu, H.; Jiao, P.; Li, S.; Wang, C., ACE2 and TMPRSS2 in human saliva can adsorb to the oral mucosal epithelium. Journal of anatomy 2022, 240, (2), 398-409.
  14. Peng, J.; Sun, J.; Zhao, J.; Deng, X.; Guo, F.; Chen, L., Age and gender differences in ACE2 and TMPRSS2 expressions in oral epithelial cells. Journal of translational medicine 2021, 19, (1), 1-11.
  15. Salas Orozco, M. F.; Niño-Martínez, N.; Martínez-Castañón, G.-A.; Patiño Marín, N.; Sámano Valencia, C.; Dipp Velázquez, F. A.; Sosa Munguía, P. d. C.; Casillas Santana, M. A., Presence of SARS-CoV-2 and Its Entry Factors in Oral Tissues and Cells: A Systematic Review. Medicina 2021, 57, (6), 1-12.
  16. Bidra, A. S.; Pelletier, J. S.; Westover, J. B.; Frank, S.; Brown, S. M.; Tessema, B., Rapid in‐vitro inactivation of severe acute respiratory syndrome coronavirus 2 (SARS‐CoV‐2) using povidone‐iodine oral antiseptic rinse. Journal of Prosthodontics 2020, 29, (6), 529-533.
  17. Herrera, D.; Serrano, J.; Roldán, S.; Sanz, M., Is the oral cavity relevant in SARS-CoV-2 pandemic? Clinical oral investigations 2020,24, (8), 2925-2930.
  18. Carrouel, F.; Gonçalves, L.; Conte, M.; Campus, G.; Fisher, J.; Fraticelli, L.; Gadea-Deschamps, E.; Ottolenghi, L.; Bourgeois, D., Antiviral activity of reagents in mouth rinses against SARS-CoV-2. Journal of dental research 2021, 100, (2), 124-132.
  19. Bidra, A. S.; Pelletier, J. S.; Westover, J. B.; Frank, S.; Brown, S. M.; Tessema, B., Comparison of in vitro inactivation of SARS CoV‐2 with hydrogen peroxide and povidone‐iodine oral antiseptic rinses. Journal of Prosthodontics 2020, 29, (7), 599-603.
  20. Peng, X.; Xu, X.; Li, Y.; Cheng, L.; Zhou, X.; Ren, B., Transmission routes of 2019-nCoV and controls in dental practice. International journal of oral science 2020, 12, (1), 1-6.
  21. Meister, T. L.; Brüggemann, Y.; Todt, D.; Conzelmann, C.; Müller, J. A.; Groß, R.; Münch, J.; Krawczyk, A.; Steinmann, J.; Steinmann, J., Virucidal efficacy of different oral rinses against severe acute respiratory syndrome coronavirus 2. The Journal of infectious diseases 2020, 222, (8), 1289-1292.
  22. Anderson, D. E.; Sivalingam, V.; Kang, A. E. Z.; Ananthanarayanan, A.; Arumugam, H.; Jenkins, T. M.; Hadjiat, Y.; Eggers, M., Povidone-iodine demonstrates rapid in vitro virucidal activity against SARS-CoV-2, the virus causing COVID-19 disease. Infectious diseases and therapy 2020, 9, (3), 669-675.
  23. Jain, A.; Grover, V.; Singh, C.; Sharma, A.; Das, D. K.; Singh, P.; Thakur, K. G.; Ringe, R. P., Chlorhexidine: An effective anticovid mouth rinse. Journal of Indian Society of Periodontology 2021, 25, (1), 86-88.
  24. Koch-Heier, J.; Hoffmann, H.; Schindler, M.; Lussi, A.; Planz, O., Inactivation of SARS-CoV-2 through Treatment with the Mouth Rinsing Solutions ViruProX® and BacterX® Pro. Microorganisms 2021, 9, (3:521), 1-9.
  25. Davies, K.; Buczkowski, H.; Welch, S. R.; Green, N.; Mawer, D.; Woodford, N.; Roberts, A. D.; Nixon, P. J.; Seymour, D. W.; Killip, M. J., Effective in vitro inactivation of SARS-CoV-2 by commercially available mouthwashes. Journal of General Virology 2021, 102, (4), 1-4.
  26. Tiong, V.; Hassandarvish, P.; Bakar, S. A.; Mohamed, N. A.; Wan Sulaiman, W. S.; Baharom, N.; Abdul Samad, F. N.; Isahak, I., The effectiveness of various gargle formulations and salt water against SARS-CoV-2. Scientific Reports 2021, 11, (1), 1-7.
  27. Meyers, C.; Robison, R.; Milici, J.; Alam, S.; Quillen, D.; Goldenberg, D.; Kass, R., Lowering the transmission and spread of human coronavirus. Journal of Medical Virology 2021, 93, (3), 1605-1612.
  28. Pelletier, J. S.; Tessema, B.; Frank, S.; Westover, J. B.; Brown, S. M.; Capriotti, J. A., Efficacy of povidone-iodine nasal and oral antiseptic preparations against severe acute respiratory syndrome-coronavirus 2 (SARS-CoV-2). Ear, Nose & Throat Journal 2021, 100, (2_suppl), 192S-196S.
  29. Rodríguez-Casanovas, H. J.; la Rosa, M. D.; Bello-Lemus, Y.; Rasperini, G.; Acosta-Hoyos, A. J., Virucidal Activity of Different Mouthwashes Using a Novel Biochemical Assay. Healthcare 2022, 10, (63), 1-10.
  30. Yoon, J. G.; Yoon, J.; Song, J. Y.; Yoon, S.-Y.; Lim, C. S.; Seong, H.; Noh, J. Y.; Cheong, H. J.; Kim, W. J., Clinical significance of a high SARS-CoV-2 viral load in the saliva. Journal of Korean medical science 2020, 35, (20), 1-6.
  31. Khan, M. M.; Parab, S. R.; Paranjape, M., Repurposing 0.5% povidone iodine solution in otorhinolaryngology practice in Covid 19 pandemic. American journal of otolaryngology 2020, 41, (5), 1-4.
  32. Seneviratne, C. J.; Balan, P.; Ko, K. K. K.; Udawatte, N. S.; Lai, D.; Ng, D. H. L.; Venkatachalam, I.; Lim, K. S.; Ling, M. L.; Oon, L., Efficacy of commercial mouth-rinses on SARS-CoV-2 viral load in saliva: randomized control trial in Singapore. Infection 2021, 49, (2), 305-311.
  33. Chaudhary, P.; Melkonyan, A.; Meethil, A.; Saraswat, S.; Hall, D. L.; Cottle, J.; Wenzel, M.; Ayouty, N.; Bense, S.; Casanova, F., Estimating salivary carriage of severe acute respiratory syndrome coronavirus 2 in nonsymptomatic people and efficacy of mouthrinse in reducing viral load: A randomized controlled trial. Journal of the American Dental Association (1939) 2021, 152, (11), 903-908.
  34. de Paula Eduardo, F.; Corrêa, L.; Heller, D.; Daep, C. A.; Benitez, C.; Malheiros, Z.; Stewart, B.; Ryan, M.; Machado, C. M.; Hamerschlak, N., Salivary SARS-CoV-2 load reduction with mouthwash use: a randomized pilot clinical trial. Heliyon 2021, 7, (6), 1-7.
  35. Huang, Y. H.; Huang, J. T., Use of chlorhexidine to eradicate oropharyngeal SARS‐CoV‐2 in COVID‐19 patients. Journal of Medical Virology 2021, 93, (7), 4370-4373.
  36. Elzein, R.; Abdel-Sater, F.; Fakhreddine, S.; Abi Hanna, P.; Feghali, R.; Hamad, H.; Ayoub, F., In vivo evaluation of the virucidal efficacy of Chlorhexidine and Povidone-iodine mouthwashes against salivary SARS-CoV-2. A randomized-controlled clinical trial. Journal of Evidence Based Dental Practice 2021, 21, (3), 1-10.
  37. Tadakamadla, J.; Boccalari, E.; Rathore, V.; Dolci, C.; Tartaglia, G. M.; Tadakamadla, S. K., In vitro studies evaluating the efficacy of mouth rinses on Sars-Cov-2: a systematic review. Journal of Infection and Public Health 2021, 14, (9), 1179-1185.
  38. Chopra, A.; Sivaraman, K.; Radhakrishnan, R.; Balakrishnan, D.; Narayana, A., Can povidone iodine gargle/mouthrinse inactivate SARS-CoV-2 and decrease the risk of nosocomial and community transmission during the COVID-19 pandemic? An evidence-based update. Japanese Dental Science Review 2021, 57, 39-45.
  39. Walsh, L. J., Safety issues relating to the use of hydrogen peroxide in dentistry. Australian dental journal 2000, 45, (4), 257-269.
  40. Hossainian, N.; Slot, D.; Afennich, F.; Van der Weijden, G., The effects of hydrogen peroxide mouthwashes on the prevention of plaque and gingival inflammation: a systematic review. International journal of dental hygiene 2011, 9, (3), 171-181.
  41. Gilbert, P.; Moore, L. E., Cationic antiseptics: diversity of action under a common epithet. J Appl Microbiol 2005, 99, (4), 703-715.
  42. Al‐Maweri, S. A.; Nassani, M. Z.; Alaizari, N.; Kalakonda, B.; Al‐Shamiri, H. M.; Alhajj, M. N.; Al‐Soneidar, W. A.; Alahmary, A. W., Efficacy of aloe vera mouthwash versus chlorhexidine on plaque and gingivitis: a systematic review. International journal of dental hygiene 2020, 18, (1), 44-51.
  43. del Río-Carbajo, L.; Vidal-Cortés, P., Tipos de antisépticos, presentaciones y normas de uso. Medicina Intensiva 2019, 43, 7-12.
  44. Haydari, M.; Bardakci, A. G.; Koldsland, O. C.; Aass, A. M.; Sandvik, L.; Preus, H. R., Comparing the effect of 0.06%-, 0.12% and 0.2% Chlorhexidine on plaque, bleeding and side effects in an experimental gingivitis model: a parallel group, double masked randomized clinical trial. BMC Oral Health 2017, 17, (1), 1-8.
  45. Kamolnarumeth, K.; Thussananutiyakul, J.; Lertchwalitanon, P.; Rungtanakiat, P.; Mathurasai, W.; Sooampon, S.; Arunyanak, S. P., Effect of mixed chlorhexidine and hydrogen peroxide mouthrinses on developing plaque and stain in gingivitis patients: a randomized clinical trial. Clinical Oral Investigations 2021, 25, (4), 1697-1704.
  46. Sato, S.; Miyake, M.; Hazama, A.; Omori, K., Povidone-iodine-induced cell death in cultured human epithelial HeLa cells and rat oral mucosal tissue. Drug and chemical toxicology 2014, 37, (3), 268-275.
  47. Shankar, S.; Saha, A.; Jamir, L.; Kakkar, R., Protection at Portal of Entry (PPE) with Povidone Iodine for COVID-19. International Journal of Medicine and Public Health 2020, 10, (4), 166-168.
  48. Hooper, S. J.; Lewis, M. A. O.; Wilson, M. J.; Williams, D. W., Antimicrobial activity of Citrox® bioflavonoid preparations against oral microorganisms. British dental journal 2011, 210, (1), 1-5.
  49. González-Mas, M. C.; Rambla, J. L.; López-Gresa, M. P.; Blázquez, M. A.; Granell, A., Volatile compounds in citrus essential oils: A comprehensive review. Frontiers in Plant Science 2019, 10, (12), 1-18.
  50. Sun, J., D-Limonene: safety and clinical applications. Alternative Medicine Review 2007, 12, (3), 259.
  51. Kampf, G.; Todt, D.; Pfaender, S.; Steinmann, E., Persistence of coronaviruses on inanimate surfaces and their inactivation with biocidal agents. J Hosp Infect 2020, 104, (3), 246-251.
  52. Koarai, A.; Sugiura, H.; Yanagisawa, S.; Ichikawa, T.; Minakata, Y.; Matsunaga, K.; Hirano, T.; Akamatsu, K.; Ichinose, M., Oxidative stress enhances toll-like receptor 3 response to double-stranded RNA in airway epithelial cells. Am J Respir Cell Mol Biol 2010, 42, (6), 651-660.
  53. Caruso, A. A.; Del Prete, A.; Lazzarino, A. I.; Capaldi, R.; Grumetto, L., Might hydrogen peroxide reduce the hospitalization rate and complications of SARS-CoV-2 infection? Infect Control Hosp Epidemiol 2020, 41, (11), 1360-1361.
  54. Komine, A.; Yamaguchi, E.; Okamoto, N.; Yamamoto, K., Virucidal activity of oral care products against SARS-CoV-2 in vitro. J Oral Maxillofac Surg Med Pathol 2021, 33, (4), 475-477.

  1. The reference should be carefully checked. The formatting is not homogenous.

References were checked

  1. There are additional numerous places throughout the manuscript where relatively minor edits in sentence structure and/or grammar would benefit the overall manuscript. This reviewer will not attempt to address all of these beyond what has already been mentioned (comments above) but it should be addressed by the authors prior to publication.

       Made English style and language 

Reviewer 2 Report

The manuscript "Oral Antiseptics against SARS-CoV-2: A Literature Review" by Bernal et al reviews the use of different oral antiseptics that could be used against SARS-CoV-2. The manuscript is interesting, written well and timely. I would like the authors to consider few points before it can be considered for publication: 

1) For the different antiseptics discussed, it would be good if the authors also discuss their side effects. 

2) So, far the manuscript is a textual description of the antiseptics used. It becomes better if some of the graphs are also presented. This is possible to do in the Introduction section. A schematic could be drawn for different antiseptics used. Or some of the results published can be taken after permission so that the readers can quickly catch the results. 

3) What are the future directions? 

Author Response

REVIEWER 2.

  1. For the different antiseptics discussed, it would be good if the authors also discuss their side effects. 

The side effects of antiseptics that have been shown to have them are discussed, however it is important to highlight that their effect occurs especially when they are used for prolonged periods, in this work we justify their use prior to the consultation dental to reduce the viral load in drops and aerosols, not prolonged use at home

6.1. Hydrogen Peroxide (H2O2)

Carrouel et al. recommended three daily mouthwashes and two nasal washes from the onset of symptoms of COVID-19, during its evolution and in the hospitalization of cases without complications[18].

Mouthwashes with a low concentration of H2O2 (≤ 1.5%) are safe, even when used over a long-term period. However, mouthwashes with a concentration of 3% H2O2 have shown adverse effects when used for 1 to 2 min[39]. These undesirable effects include a mouth pain sensation, burning sensation, erythema, edema, ulcers, and/or erosive changes in the oral mucosa[40].

6.2. Chlorhexidine (CHX)

The most commonly observed side effects associated with the long-term use of 0.06%, 0.12%, and 0.2% concentrations of CHX include a loss of taste, numbness, burning sensation, oral mucosal pain, (including tongue and gums), dryness, hypersensitivity reactions, or photosensitivity[42-45].

6.3. Povidone–iodine (PVP-I)

The cytotoxic effects and tolerance of PVP-I are important points to consider for its implementation, because it is toxic to the oral and nasal mucosa at concentrations above 2.5% and 5%, respectively, although commercial formulations do not reach these concentrations[17]. Furthermore, it can be safely administered for five months in the nasal cavity and six months in the oral cavity. Prolonged use of PVP-I in concentrations of 1% to 1.25% does not irritate the mucous membranes or produce adverse effects for up to 28 months, nor does it stain the teeth or alter taste functions [46, 47]. Its implementation has not been shown to affect thyroid function, but an increase in serum thyroid-stimulating hormone has been observed in individuals undergoing prolonged PVP-I treatment (24 weeks)[38].

  1. So, far the manuscript is a textual description of the antiseptics used. It becomes better if some of the graphs are also presented. This is possible to do in the Introduction section. A schematic could be drawn for different antiseptics used. Or some of the results published can be taken after permission so that the readers can quickly catch the results. 

Added a figure of the mechanisms of action of oral antiseptics

Figure 1. Mechanism of oral antiseptics. Hydrogen peroxide (H2O2) produces hydroxyl free radicals and reactive oxygen species that react with lipids, proteins, and RNA. Povidone–iodine (PVP-I) oxides -SH groups to –S-S- and -NH2groups to -NO2. Chlorhexidine (CHX) binds to membrane phospholipids and displaces viral protein cations by CHX anion exchange. Cetylpyridinium chloride (CPC) displaces cations and neutralizes negative -COO- charges of proteins, which breaks the viral membrane.

  1. What are the future directions? 

Future directions are outlined in the conclusion.

Adopting these simple preventive measures, oral health professionals can reduce their risk of infection by reducing the viral load in the mouth of patients. However, it is important to highlight that the impact of a "false sense of security" on health professionals and patients and society should not be underestimated, because this can lead to a reduction in the use of protective equipment or closer social interactions with potentially infected people, thereby increasing SARS-CoV-2 infections. Currently, clinical trials and in vivo studies on the virucidal effect against SARS-CoV-2 are limited; thus, more research of this type is needed, in order to implement protocols or clinical practice guidelines against SARS-CoV-2 or other emerging microorganisms. In addition, it is necessary for health professionals to be up to date in infection control, even more so in pandemics such as COVID-19, where the use of antiseptics could be enormous help in preventing the spread of SARS-CoV-2.

Made English style and language

Reviewer 3 Report

In this manuscript, several concerns need to be addressed as follows:

1. Many formatting errors exist especially the capitalization of the letters. E.g. 27 Health Institutions should not be capitalized.

2. The introduction should end with the aim of the review and outline the presented points.

3. It is not preferable to begin sentences with abbreviations like that in lines 124 H2O2…..etc.

4. All the types of oral antiseptic presented should be outlined as subheadings of a general heading (E.g. 6. types of oral antiseptics against SARS-CoV-2 then 6.1. Hydrogen peroxide…and so on).

5. The references throughout the manuscript need to be revised. Here are some examples:

- Line 174-176: Jain et al., found that its viricidal effect was the same at 30 and 60 s of exposure, but that it was slightly higher if it was used at 175 0.2% concentration [22, 29]. One reference for the study of Jain et al., should be cited. However, the authors cited two references.

- Line 179-180: Yoon et al., evaluated the effect on salivary the viral load after gargling 179 at 1, 2, and 4 hours with a chx rinse; viral load was low at 2 h after gargling, but then 180 increased [18, 30]. One reference for the study of Yoon et al., should be cited. What is the aim of citing reference no. 18.

6. A schematic figure outlining the different mechanisms of action of different types of oral antiseptics against SARS-CoV-2 is highly recommended to be added.

7. Also, I would recommend the authors give more discussions on lessons learned from the state of the science and challenges in this field in a new separate discussion section, to show the manuscript's contribution more clearly. 

Author Response

REVIEWER 3

  1. Many formatting errors exist especially the capitalization of the letters. E.g. 27 Health Institutions should not be capitalized.

Formatting errors were reviewed and changed in the document, they are marked with yellow

Example:

Abstract: Dentists are health care workers with the highest risk of exposure to COVID-19, because the oral cavity is considered to be a reservoir for SARS-CoV-2 transmission. The identification of SARS-CoV-2 in saliva, the generation of aerosols, and the proximity to patients during dental procedures are conditions that have led to these health care workers implementing additional disinfection strategies for their protection. Oral antiseptics are widely used chemical substances due to their ability to reduce the number of microorganisms. Although there is still no evidence that they can prevent the transmission of SARS-CoV-2, some preoperative oral antiseptics have been recommended as control measures, by different health institutions worldwide, to reduce the number of microorganisms in aerosols and droplets during dental procedures. Therefore, this review presents the current recommendations for the use of oral antiseptics against SARS-CoV-2 and analyzes the different oral antiseptic options used in dentistry.

Keywords: COVID-19; SARS-CoV-2; oral antiseptic; infection control

1. Introduction

Coronavirus disease 19 (COVID-19), responsible for the pandemic that began in late 2019 in Wuhan, China, is caused by a virus that originated as a new type-2 severe acute respiratory syndrome coronavirus (SARS-CoV-2), and there have been more than 536 million confirmed cases up to June 2022 [1, 2]; the main transmission route is through direct contact or the inhalation of aerosols[3]. The Occupational Safety and Health Administration (OSHA) of the United States Department of Labor established an occupational risk pyramid for COVID-19, which is structured into four levels of risk exposure: very high, high, medium, and low. Health workers are considered a group with a very high risk of exposure, because their procedures can generate aerosols; this particularly includes dentists, due to the close contact with patients and the instruments used[4].

  1. The introduction should end with the aim of the review and outline the presented points.

The aim of the work was written

Thus, the aim of the present study was to evaluate some of the antiseptic agents most frequently used by dental professionals and their antiviral effects against SARS-CoV-2 from both in vitro and in vivo studies to identify the most common compounds effective in reducing the viral load in the oral cavity in people infected with the virus.

  1. It is not preferable to begin sentences with abbreviations like that in lines 124 H2O2…..etc.
  2. All the types of oral antiseptic presented should be outlined as subheadings of a general heading (E.g. 6. types of oral antiseptics against SARS-CoV-2 then 6.1. Hydrogen peroxide…and so on).

Avoid starting with abbreviations and rinses were described as subheadings in a header

Example:

6. Types of oral antiseptics against SARS-CoV-2

6.1. Hydrogen Peroxide (H2O2)

Hydrogen peroxide is a widely used chemical compound with antimicrobial properties, and its efficacy has been demonstrated on different human viruses, including influenza and coronavirus viruses. It mainly affects viruses with a lipid envelope, such as SARS-CoV-2, through the generation of oxygen free radicals (Figure 1) [18]. One of the first strategies implemented to prevent SARS-CoV-2 transmission during dental procedures was the use of a preoperative rinse consisting of 1% hydrogen peroxide solution to reduce viral load, due to the vulnerability of the virus to oxidation[3]. In vitro studies concluded that clinically recommended and commercially available concentrations of 1.5% and 3.0% H2O2 rinses showed minimal virucidal activity at 15 s and nearly the same effect at 30 s after application[19] (Table 1). Other in vitro studies compared the virucidal effect of a mouthwash with a concentration of 1.5% H2O2 versus mouthwashes with other active ingredients (povidone–iodine [PVP-I], chlorhexidine [CHX], ethanol+essential oils) at 30 s of exposure, showing that the virucidal activity of H2O2 was less effective[21, 37]. However, Xu et al. reported a >99.9% clearance rate of SARS-CoV-2 after a contact time of 30 min [2](Table 1). Carrouel et al. recommended three daily mouthwashes and two nasal washes from the onset of symptoms of COVID-19, during its evolution and in the hospitalization of cases without complications[18]. This measure, interestingly, could be applied to hospitalized patients, because the risk of transmission is increased in procedures such as ventilation, intubation, and non-invasive aspiration, which generate bioaerosols and result in nosocomial transmission from hospitalized infected people to healthy family members, caregivers, health professionals, and other patients in the hospital[38]. In a randomized controlled clinical trial, Chaudhary et al. showed that a mouth rinse with 1% H2O2 achieved mean viral load reductions of 61% to 89% at 15 min after application, and 70% to 97% at 45 min[33] (Table 2). In this sense, Eduardo et al. demonstrated that a 1.5% H2O2 rinse maintained a significant decrease in SARS-CoV-2 viral load up to 30 min after application, as compared with mouthwashes containing 0.075% cetylpyridinium chloride (CPC) + 0.28% zinc lactate and 0.12% chlorhexidine, which maintained a significant decrease in viral load up to 60 min after use[34] (Table 2). Meyers et al. evaluated various nasal and oral rinses in vitro, and tested three mouthwashes that included 1.5% H2O2 as the active ingredient which showed similar abilities to inactivate SARS-CoV-2, with reductions of viral load of <90% to 99% at 30 s during contact time and >90% to <99.99% at 1 min of exposure[27] (Table 1). However, not all studies have shown a reduction in viral load after using H2O2 rinses. In their clinical study, Gottsauner et al. concluded that a mouth rinse with 1% H2O2 does not reduce the intraoral viral load in subjects positive for SARS-CoV-2 (Table 2); therefore, additional studies are needed to evaluate the use of mouth rinses containing other active agents to reduce the intraoral load of SARS-CoV-2[3]. Mouthwashes with a low concentration of H2O2 (≤ 1.5%) are safe, even when used over a long-term period. However, mouthwashes with a concentration of 3% H2O2 have shown adverse effects when used for 1 to 2 min[39]. These undesirable effects include a mouth pain sensation, burning sensation, erythema, edema, ulcers, and/or erosive changes in the oral mucosa[40].

6.2. Chlorhexidine (CHX)

        CHX is an antiseptic and biguanide disinfectant, with widely demonstrated antimicrobial activity against Gram-positive, Gram-negative, anaerobic, and aerobic bacteria, as well as some viruses and yeasts[17]. The mechanism of action of the biguanides is based on the strong association of the biguanide group with the anions exposed in the membrane and cell wall of the microorganism, particularly acid phospholipids and proteins (Figure 1) [41]. This is considered the gold standard for biofilm control, and its side effects are well known. CHX can be effective against enveloped viruses (herpes simplex virus, HIV, influenza virus, cytomegalovirus), but not against small non-enveloped viruses (enterovirus, poliovirus, human papillomavirus)[17]; in this context, it could also be effective against SARS-CoV-2[5]. One study evaluated its effectiveness against SARS-CoV-2 at concentrations of 0.2% and 0.12% (Table 1). Its viricidal effect at 30 s was lower than povidone–iodine (PVP-I)[23, 37]. Jain et al. found that its viricidal effect was the same at 30 and 60 s of exposure, although it was slightly higher if used at a 0.2% concentration [23] (Table 1). Xu et al. carried out an in vitro study where they observed that the potency of CHX was greater when the product was not washed off after virus binding, suggesting that the prolonged effect of rinses on cells affects antiviral outcome [2] (Table 1). Yoon et al. evaluated the effect on the salivary viral load after gargling at 1, 2, and 4 hours with a CHX rinse; viral load was low at 2 h after gargling, but then increased [30] (Table 2). A randomized controlled clinical trial by Chaudhary et al. showed that a CHX rinse at 0.12% over 60 s achieved a mean reduction of 61% to 89% of viral load after 15 min, and a reduction of 70% to 97% after 45 min[33] (Table 2). Huang et al. concluded that in patients who combined the use of a mouth rinse and oropharyngeal spray with 0.12% CHX, they eliminated SARS‐CoV‐2 in 86% compared with 6.3% of a control group[35] (Table 2). The most commonly observed side effects associated with the long-term use of 0.06%, 0.12%, and 0.2% concentrations of CHX include a loss of taste, numbness, burning sensation, oral mucosal pain, (including tongue and gums), dryness, hypersensitivity reactions, or photosensitivity[42-45]. Heidari et al. determined that loss of taste is significantly greater in 0.2% CHX compared with concentrations at 0.12% and 0.06%, from the seventh day of use[44]. In addition, extrinsic brown stains may appear on teeth, dentures, composite restorations, and the tongue. The severity of dental stains can vary between patients and worsen when people also consume tea, port, red wine, and other tannin-containing substances[45].

6.3. Povidone–iodine (PVP-I)

PVP-I is an oxidizing agent composed of iodine and the water-soluble polymer polyvinylpyrrolidone. It exerts antimicrobial activity when it dissociates and releases iodine, which penetrates and alters protein synthesis, oxidizes nucleic acids, and lyses bacteria, fungi, and viruses (Figure 1) [18]. The recommendation of using an oral antiseptic with PVP-I in patients with COVID-19 is based on its virucidal activity against enveloped and non-enveloped viruses, including Ebola, Middle East respiratory syndrome (MERS), coronavirus (SARS), influenza, and hand, foot, and mouth disease (Coxsackievirus)[17].

The different commercial presentations of PVP-I, which include antiseptic solution (10%), hand sanitizer (7.5%), throat spray (0.45%), and mouthwash (1%), show reductions in the viral load of SARS-CoV-2, with a clearance rate of >99.99% after an exposure time of 30 s[22] (Table 1). Meyers et al. evaluated various nasal and oral rinses in vitro and concluded that a 5% PVP-I solution inactivates SARS-CoV-2 by >90% to <99.9% in a minimum contact time of 30 s, unlike three other mouth rinses that contain 1.5% H2O2, which reached the same inactivation at 1 min of contact time[27] (Table 1). Pelletier et al. performed an in vitro study with three nasal and three oral antiseptics with different concentrations of PVP-I; they concluded that all presentations tested were effective in inactivating SARS-CoV-2 after 60 s of exposure[28] (Table 1). Another study concluded that PVP-I at concentrations of 0.5%, 1%, and 1.5% completely inactivated SARS-CoV-2 within 15 s of contact[16] (Table 1). In comparison with H2O2, PVP-I exhibited a better virucidal activity at 15 s of exposure[19] (up to a threefold greater effect) (Table 1). Regarding clinical trials, Chaudhary et al. showed that a 0.5% PVP-I rinse for 60 s achieved a mean reduction of 61% to 89% in the viral load after 15 min, and a reduction of 70% to 97% after 45 min [33] (Table 2). Elzein et al. conducted a randomized controlled clinical trial which demonstrated that a mouth rinse with 1% PVP-I significantly reduced the intraoral viral load in subjects positive for SARS-CoV-2 [36](Table 2). Khan et al. recommended the application of 0.5% PVP-I nasal drops in addition to 0.5% PVP-I mouth rinses for 30 s to achieve a reduction in SARS-CoV-2 viral load. However, the period of time during which antisepsis remains needs to be investigated through several randomized controlled studies[31] (Table 2).

The cytotoxic effects and tolerance of PVP-I are important points to consider for its implementation, because it is toxic to the oral and nasal mucosa at concentrations above 2.5% and 5%, respectively, although commercial formulations do not reach these concentrations[17]. Furthermore, it can be safely administered for five months in the nasal cavity and six months in the oral cavity. Prolonged use of PVP-I in concentrations of 1% to 1.25% does not irritate the mucous membranes or produce adverse effects for up to 28 months, nor does it stain the teeth or alter taste functions [46, 47]. Its implementation has not been shown to affect thyroid function, but an increase in serum thyroid-stimulating hormone has been observed in individuals undergoing prolonged PVP-I treatment (24 weeks)[38]. Contraindications should be considered in patients with an anaphylactic allergy to iodine, pregnancy, active thyroid disease, and patients receiving radioactive iodine therapy. The alternative to PVP-I oral solution is the use of a 1.5% H2O2 rinse, as recommended by interim ADA guidelines[16].

  1. The references throughout the manuscript need to be revised. Here are some examples:

References were checked throughout the document

  1. A schematic figure outlining the different mechanisms of action of different types of oral antiseptics against SARS-CoV-2 is highly recommended to be added.

Added a figure of the mechanisms of action of oral antiseptics

Figure 1. Mechanism of oral antiseptics. Hydrogen peroxide (H2O2) produces hydroxyl free radicals and reactive oxygen species that react with lipids, proteins, and RNA. Povidone–iodine (PVP-I) oxides -SH groups to –S-S- and -NH2 groups to -NO2. Chlorhexidine (CHX) binds to membrane phospholipids and displaces viral protein cations by CHX anion exchange. Cetylpyridinium chloride (CPC) displaces cations and neutralizes negative -COO- charges of proteins, which breaks the viral membrane.

  1. Also, I would recommend the authors give more discussions on lessons learned from the state of the science and challenges in this field in a new separate discussion section, to show the manuscript's contribution more clearly. 

Added a discussion section

Discussion

Dentists experience the highest risk of exposure to COVID-19 infection due to the dental devices and instruments used (ultrasound, handpieces, triple syringes, etc.) that can generate large amounts of aerosols, which disperse numerous bacteria and viruses[5]. The presence of SARS-CoV-2 has been identified in different types of materials, remaining in plastic for up to 4 days, and 7 days in stainless steel and surgical masks[8]; for this reason, additional measures are being sought that could reduce the viral load of SARS-CoV-2 and thereby reduce its transmission. Oral rinses could significantly reduce the viral load because saliva is one of the main vectors. Coughing once or having a five-minute conversation can produce up to 3,000 droplets of saliva, whereas a sneeze can produce up to 40,000 droplets [18]. It has recently been shown that mouthwashes can rapidly inactivate SARS-CoV-2 through in vitro [2, 16, 19, 21-29] and in vivostudies [3, 30-36]. Subsequently, there have been extensive discussions regarding the utilization of mouth rinses to possibly complement current prevention measures such as facemasks, hand disinfection, and social distancing in order to reduce the global spread of SARS-CoV-2[18]. To identify the effect of oral antiseptics, we conducted a literature review of the active ingredients that have demonstrated effects on SARS-CoV-2 both in vitro and in vivo. The efficient inactivation of coronaviruses (SARS and MERS) on inanimate surfaces using hydrogen peroxide (0.5% H2O2 for 1 minute) was evaluated by Kampf et al., demonstrating good results [51]. H2O2 application was a one of the first strategies implemented to prevent SARS-CoV-2 transmission during dental procedures is the use of a preoperative rinse consisting of 1% hydrogen peroxide solution to reduce viral load, due to the vulnerability of the virus to oxidation[3]. Meyers et al. evaluated various nasal and oral rinses in vitro, and tested three mouthwashes that included 1.5% H2O2 as the active ingredient that showed similar abilities to inactivate SARS-CoV-2 with a reduction in viral load of <90% to 99% at 30 s during a contact time and >90% to <99.99% at 1 min of exposure[27]. For this reason, it has been proposed that hydrogen peroxide, as an antiseptic agent, could play a fundamental role in reducing the rate of hospitalization and complications associated with COVID-19. The antiseptic action is due not only to the known oxidative and mechanical scavenging properties of hydrogen peroxide, but also to the induction of the innate antiviral inflammatory response through overexpression of Toll-like receptor 3 (TLR3) [52]. No damage to the oral mucous membranes or their microvilli was observed after continuous treatment with 3% H2O2 gargling[53].

CHX is considered to be the gold standard for biofilm control; it can be effective against enveloped viruses (such as herpes simplex virus, HIV, influenza virus, and cytomegalovirus), but not against small non-enveloped viruses (such as enterovirus, poliovirus, and human papillomavirus)[17]; in this context, it could also be effective against SARS-CoV-2[5]. A high level of virus in saliva was detected in a clinical trial performed by Yoon et al. in 2020, but CHX was able to significantly decrease the viral load for 2 h after a single use [30].

Huang et al. concluded that in patients who combined the use of a mouth rinse and oropharyngeal spray with 0.12% CHX, they eliminated SARS‐CoV‐2 in 86% of subjects compared with 6.3% of a control group[35]. In summary, it is concluded that CHX may exert an interesting virucidal efficacy against HSV-1 and influenza A viruses. However, reductions in SARS-CoV-2 strains have not yet been demonstrated when evaluated in vitro. The use of a CHX mouthwash was identified to temporarily reduce the viral load of SARS-CoV-2 in patients with COVID-19 [30, 35].

Regarding PVP-I, Meyers et al. evaluated various nasal and oral rinses in vitro and concluded that a 5% PVP-I solution inactivates SARS-CoV-2 by >90% to <99.9% after a minimum contact time of 30 s, unlike three other mouth rinses containing 1.5% H2O2, which reached the same inactivation at 1 min of contact time[27]. Elzein et al. conducted a randomized controlled clinical trial which demonstrated that a mouth rinse with 1% PVP-I significantly reduced the intraoral viral load in subjects positive for SARS-CoV-2 [36]. However, the period of time during which antisepsis remains needs to be investigated through several randomized controlled studies [31].

Komine et al. also showed that mouth rinses containing 0.04–0.075% CPC inactivated >99.99% of SARS-CoV-2 in 20–30 s [54].

Koch-Heier et al. evaluated two rinses—0.05% CPC and 1.5% H2O2, and 0.1% CHX, 0.05% CPC, and 0.005% fluoride—and their results showed that rinses with 0.1% CHX and 1.5% H2O2 did not result in a reduction in viral load, but that 0.05% CPC present in both rinses was responsible for the virucidal effect against SARS-CoV-2, which significantly reduced the viral load [24].

Although CPC has demonstrated antiviral activity against several viruses that cause respiratory infections, more research is needed to elucidate the action of CPC against SARS-CoV-2[18]. Rodríguez-Casanovas et al. evaluated the virucidal activity of different mouthwashes, and found that D-limonene resulted in a significant reduction in virucidal activity of around 99.99% against SARS-CoV-2, through a solution containing D-limonene (0.2%) and CPC (0.05%)[29].

Made English style and language changes

Round 2

Reviewer 2 Report

The authors have revised the manuscript and it looks better than the initial submission.

A few minor comments before it can be published in the journal:

A/C to the description included in the manuscript, hydrogen peroxide is found to be very effective in inhibiting SARS-CoV2 infectivity. In recent years, there has been increasing interest in the use of cold plasmas and they have also been applied against SARS-CoV2 disinfection . This is due to their potential to generate hydrogen peroxide  at atmospheric conditions.

I suggest the authors to kindly also review this and if relevant add this. If that is, the readers of the article can go to a very broad area.

Author Response

A/C to the description included in the manuscript, hydrogen peroxide is found to be very effective in inhibiting SARS-CoV2 infectivity. In recent years, there has been increasing interest in the use of cold plasmas and they have also been applied against SARS-CoV2 disinfection . This is due to their potential to generate hydrogen peroxide  at atmospheric conditions.

 I suggest the authors to kindly also review this and if relevant add this. If that is, the readers of the article can go to a very broad area.

Added information on the use of cold plasmas as a disinfection method against SARS-CoV2.

Some health institutions worldwide have suggested the use of disinfection methods to eliminate SARS-CoV-2 based on H2O2. Cold atmospheric plasma technology is a method that generates a high concentration of H2O2 that causes oxidation of amino acids, nucleic acids, and induces peroxidation of unsaturated fatty acid through interaction with membrane lipids, altering the function of membranes in microorganisms. Chen et al. found that cold at-mospheric plasma eliminated SARS‐CoV‐2 on the surface of living organisms within 180 s [37]. In addition to its potential benefits in infection control and wound healing, cold atmospheric plasma has also been applied in studies targeting hemostasis control, treatment of skin diseases, immunotherapy, and regenerative medicine [37].

Reviewer 3 Report

The authors adequately responded to all comments and performed the required modifications.

Author Response

The authors adequately responded to all comments and performed the required modifications.

Thanks for your valuable suggestions